Journal of Machine Learning Research 1 (2000) 1-48    Submitted 4/00; Published 10/00

# Automatic and explainable grading of meningiomas from histopathology images

**Jonathan Ganz**                                        JONATHAN.GANZ@THI.DE
*Technische Hochschule Ingolstadt, Ingolstadt, Germany*

**Tobias Kirsch**
*Institute of Neuropathology, University Hospital Erlangen, Erlangen,*
*Germany*

**Lucas Hoffmann**
*Institute of Neuropathology, University Hospital Erlangen, Erlangen,*
*Germany*

**Christof A. Bertram**
*Institute of Pathology, University of Veterinary Medicine Vienna, Vienna, Austria*

**Christoph Hoffmann**
*Technische Hochschule Ingolstadt, Ingolstadt, Germany*

**Andreas Maier**
*Pattern Recognition Lab, Computer Science, Friedrich-Alexander-Universität Erlangen-Nürnberg,*
*Erlangen, Germany*

**Katharina Breininger**
*Department Artificial Intelligence in Biomedical Engineering, Friedrich-Alexander-Universität Erlangen-*
*Nürnberg, Erlangen, Germany*

**Ingmar Blümcke**
*Institute of Neuropathology, University Hospital Erlangen, Erlangen,*
*Germany*

**Samir Jabari**
*Institute of Neuropathology, University Hospital Erlangen, Erlangen,*
*Germany*

**Marc Aubreville**
*Technische Hochschule Ingolstadt, Ingolstadt, Germany*

**Editor:**

## Abstract

Meningioma is one of the most prevalent brain tumors in adults. To determine its malignancy, it is graded by a pathologist into three grades according to WHO standards. This grade plays a decisive role in treatment, and yet may be subject to inter-rater discordance. In this work, we present and compare three approaches towards fully automatic meningioma grading from histology whole slide images. All approaches are following a two-stage paradigm, where we first identify a region of interest based on the detection of mitotic figures in the slide using a state-of-the-art object detection deep learning network. This region of highest mitotic rate is considered characteristic for biological tumor behavior. In

the second stage, we calculate a score corresponding to tumor malignancy based on information contained in this region using three different settings. In a first approach, image patches are sampled from this region and regression is based on morphological features encoded by a ResNet-based network. We compare this to learning a logistic regression from the determined mitotic count, an approach which is easily traceable and explainable. Lastly, we combine both approaches in a single network. We trained the pipeline on 951 slides from 341 patients and evaluated them on a separate set of 141 slides from 43 patients. All approaches yield a high correlation to the WHO grade. The logistic regression and the combined approach had the best results in our experiments, yielding correct predictions in 32 and 33 of all cases, respectively, with the image-based approach only predicting 25 cases correctly. Spearman's correlation was 0.7163, 0.7926 and 0.7900 respectively. It might be counter-intuitive at first that morphological features provided by the image patches do not improve model performance. Yet, this mirrors the criteria of the grading scheme, where mitotic count is the only unequivocal parameter.

**Keywords:** automatic tumor grading, meningioma, deep learning, known operator learning

## 1. Introduction

With 20-30% of all primary brain tumors, meningiomas are reported to be the most frequent occurring brain tumor in adults (Lam Shin Cheung et al. (2018), Saraf et al. (2011)). Meningiomas are classified into various sub-types, and graded according to the grading system of the World Health Organization (WHO) into three grades with ascending risk of recurrence and/or aggressive growth (Louis et al. (2016)). Grade I is the most prevalent, accounting for 80 to 90% of all meningiomas. However, even though these low-grade tumors are mostly benign, recurrence rates range from 7 to 20% (Louis et al. (2016)). Grade II and III meningiomas are less frequently diagnosed, but tend to show a more aggressive biological behavior than grade I meningiomas (Louis et al. (2016)). For these, recurrence rates are reported to be in the range of 30 to 40% for grade II and 50 to 80% for grade III meningiomas (Louis et al. (2016)). These differences make grading an important factor for treatment success and tumor management, however, concordance between raters was reported to be suboptimal (Rogers et al. (2015)).

Beside morphological features like high cellularity, prominent nucleoli or brain invasion, the presence of cells undergoing cell division (mitotic figures) is a key factor in the WHO grading scheme (Louis et al. (2016)). Even though mitoses are also part of tumor morphology, their density (mitotic rate) is still treated as a separate factor in the WHO grading scheme and is known to be highly correlated with cell proliferation, which is a key predictor for biological tumor behaviour (Baak et al. (2009)). Consequently, the rate of mitoses per area (mitotic count, MC), typically counted over ten high power fields, is a factor in many grading schemes, e.g. for breast cancer (Elston and Ellis (1991)) or lung cancer Kadota et al. (2012). Yet, it is also known that the inter-rater agreement on mitotic figures is fairly modest, and that algorithmic approaches offer performance in mitotic figure detection comparable to humans (Meyer et al. (2005), Malon et al. (2012), Veta et al. (2016)), Aubreville et al. (2020)).

In this work we perform automated grading of meningiomas from whole slide images (WSIs), based on deep learning models. We base our approaches on the prediction of mitotic figures by a state-of-the-art deep learning architecture. Over all images associated with one patient,

we calculate the highest mitotic count (i.e., the number of mitotic figures per area equivalent to 10 high power fields). This prediction is then used in three different approaches: First, we evaluate the performance of a model based on a pre-trained ResNet18 stem to regress the WHO grade solely based on histopathology patches. Second, the predicted mitotic count alone is used to train a very simple network to map the WHO grade. Finally, we combine both approaches and derive a novel explainable model architecture that makes use of the mitotic count as well as image information, mimicking the diagnostic procedure described in the WHO grading scheme.

## 2. Related Work

Several authors addressed the preoperative grading of meningiomas in magnetic resonance imaging (Zhang et al. (2020), Yan et al. (2017), Lin et al. (2019)). For other tumor types, automatic grading is an active field of research. For grading prostate cancer, the sum of the two most common Gleason patterns, called Gleason score is used. The score is a measure for glandular separation and thus, cancer aggressiveness (Nguyen et al. (2017)). There have been multiple attempts to asses the Gleason grade via algorithmic approaches (Nguyen et al. (2017), Lucas et al. (2019)); however, the proposed approaches are of limited transferability since mitotic figures do not play a role in Gleason grading. A more similar application is the determination of proliferation scores of breast cancer tissue, where mitotic count is an important predictive biomarker (Van Diest et al. (2004)). As with meningiomas, the density of mitotic figures is a criterion for determining tumor proliferation. In the TUPAC16 challenge, participants were faced with the tasks of predicting mitotic scores as well as the gene expression-based PAM50 score from WSIs of breast cancer tissue (Veta et al. (2019)). The solutions proposed by the participants can be dived into two groups. The one group identified a region of interest (ROI) in which they detected mitotic figures. The second group also detected a ROI but tried to predict tumor proliferation directly (Veta et al. (2019)). A key difference between these works submitted in the TUPAC16 challenge and ours is that we aim for a tumor severity prediction instead of only predicting proliferation scores. Shah et al. also targeted the prediction of tumor proliferation for breast cancer WSIs (Shah et al. (2017)). In their work, they used a pipeline of different networks to use mitotic figures as well as general morphological features from histopathological slides to aggregate a categorical tumor grade and RNA expression predictions (Shah et al. (2017)). Their approach is related to ours as we also combine the mitotic count with general morphological features. As one key difference to their approach, our model is designed to be as simple as possible and thus explainable in the contributions of the pipeline elements. Besides, to the authors' best knowledge, this is the first time automatic meningioma grading of histopathology whole slide images was performed.

## 3. Materials and Methods

### 3.1 Datasets

For this work, three different datasets were used. For all of them, hematoxylin and eosin (H&E)-stained meningioma samples were retrospectively collected from the Department of

Neuropathology, University Hospital Erlangen, Germany. All samples were digitized using an Hamamatsu S60 digital slide scanner.

- The **training dataset for mitotic figure detection** consists of 65 WSIs, completely annotated for mitotic figures. For the annotation, the WSIs were screened for mitotic figures and mitotic figure lookalikes by an expert (TK) in mitosis detection using an open source software solution (Aubreville et al. (2018)). Additionally, to avoid missed mitotic figures, a machine learning system was trained in a cross-validation scheme to find additional mitotic figures with high sensitivity, following the procedure described in (Bertram et al. (2020)). All newly found candidates were then re-evaluated by the expert and classified into being a mitotic figure or not. In total, 178,826 cell annotations were generated by this procedure. All annotations were subsequently assessed blindly (without knowing the first expert's class label) by a pathologist with five years of experience in histopathology and mitotic figure identification (CB). Disagreed cases were (again blindly) re-evaluated by a third expert, who is a trained neuropathologist (SJ). Overall, the data set contains 10,662 annotations for mitotic figures and 168,164 annotations for non-mitotic cells.

- For the **meningioma grading training dataset**, H&E stained tissue slides of the years 2009 until 2011 were collected from the hospital's slide archive. All samples were reviewed by an expert neuropathologist. Samples without sufficient tissue or with pale stains were excluded from the study. The original selection contained 47 additional WSIs which were excluded due to a possible case bleed to the test set. After this process, 951 samples / whole slide images were included in the study, representing tumor sections from 341 patients with corresponding tumor grades. For each patient, the overall WHO grade of the tumor was retrieved from the hospital information system, leading thus to 341 assigned tumor grades (272 samples of WHO grade 1, 62 samples of WHO grade 2 and 7 samples of WHO grade 3). We would like to highlight that the retrospective data collection may have resulted in some inconsistencies in the associated labels. The tumor grade was derived from patient records based on the most malignant WSI sample. This sample, however, may not be present in the dataset at hand (which is from a restricted time range, as stated). This kind of label noise can be tolerated in the training set from our point of view if utmost case is taken for the curation of the test set as described next.

- The **independent test set** consists of 121 WSIs from 43 tumor cases, representing 26 patients which are neither part of the mitotic figure detection dataset nor of the meningioma grading training dataset (17 samples of WHO grade 1, 17 samples of WHO grade 2 and 9 samples of WHO grade 3). For each WSI, a neuropathologist re-evaluated the WSIs to contain a sufficient amount of tumor tissue and confirmed sufficient scanning and staining quality. The dataset represents a complete list of samples from each of the 26 patients, ranging from 2003 to 2013. For each tumor sample, a WHO grading was performed by an expert neuropathologist, thus 43 grades were assigned within the complete set.

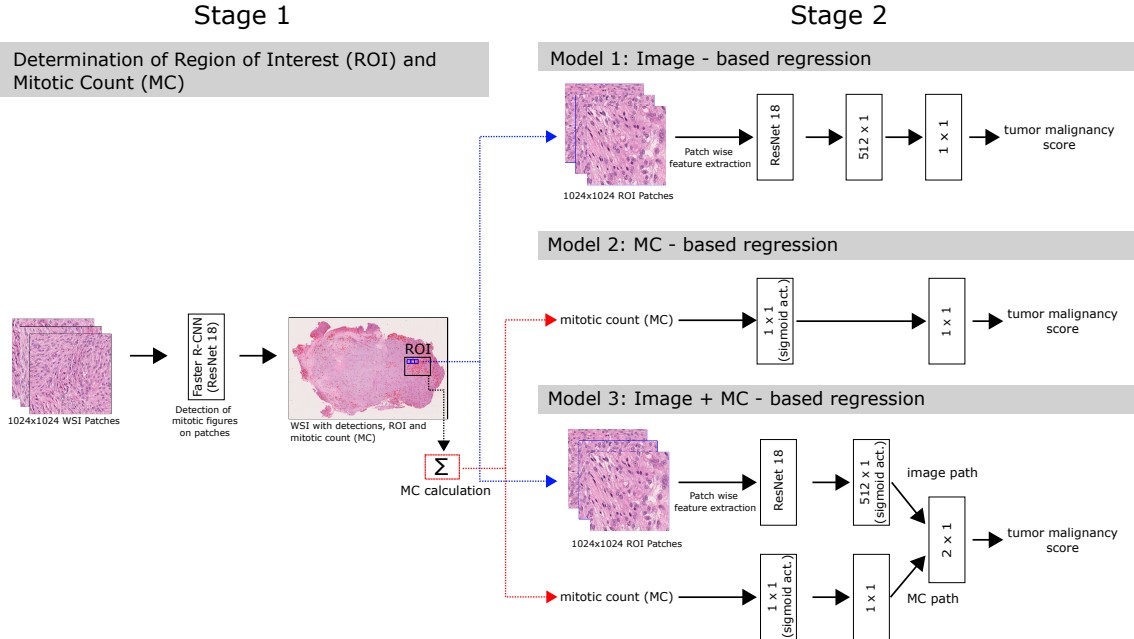

Figure 1: Overview of the three approaches compared in this work. All are based on a initial mitotic figure (MF) detection to select the region of interest (ROI) and/or to calculate the mitotic count (MC), i.e. the MF within the ROI.

## 3.2 Methods

The aim of this work is to predict a tumor malignancy score for meningiomas given a WSI as an input. In contrast to the WHO grade, in this study we used a continuous score to determine tumor malignancy. The aim is to show a smoother transition between different malignancy levels than is possible with a discrete scale like the WHO grade. Our method consists of two main stages (see Figure 1). In the first stage, mitotic figures are detected with a state-of-the-art object detector. Here we used a Faster R-CNN with a ResNet18 as backbone (Ren et al. (2017); He et al. (2016)). A mitotic figure was defined to be a square bounding box annotation with width and height of 50 pixels (approx. 2.43 $\mu m^2$). We used the aforementioned training dataset for mitotic figure detection on which we performed a random split on whole slide image level, leading to a train, validation and test set of 34, 10 and 21 WSIs, respectively. We trained the model until convergence, as observed by the validation loss. Selection of the best model parameters was performed retrospectively based on the minimal validation loss. During inference, the whole WSI was fed patch-wise into the detector. For this, adjacent patches were cut out of the WSI with an overlap of 10%. After model inference, the detections were projected back onto the WSI and overlapping detections were filtered by a non-maximum suppression (NMS) algorithm. We optimized the detection threshold on inference results on the complete training set WSIs, using the F1 score as metric. We then ran inference on the test set and subsequently estimated the mitotic count (MC) from the detected figures using a moving window average with a size

equivalent to $2.5mm^2$ (approximately 10 high power fields) at an aspect ratio of 4:3. The maximum MC determined the ROI for the subsequent second model stage (see Figure 1). This methodology is in line with the grading standard, in that the most mitotically active region is assumed to be of the highest prognostic value for tumor behavior.

In the second stage, the calculated values were used to determine a malignancy score as a continuous value (regression). In this study, we compared three different approaches to do this regression. The first approach, denoted as model 1 is a purely image based one, based on patches sampled from the ROI, which is, as indicated, assumed to determine biological tumor behavior and thus can be assumed to also contain discriminative morphological information. The ROI is fed patch-wise into a neural network based on a pre-trained ResNet18 stem and followed by a $512 \times 1$ linear layer trained from scratch. This way, the features calculated by the ResNet are used to calculate a tumor malignancy score directly from the patches. The ResNet used was pre-trained with ImageNet and fine-tuned with patches from the meningioma grading training dataset. Since a separate value is calculated for each patch of the ROI during inference, we need to combine these values into a single overall tumor malignancy score, for which we use averaging of all single values. We use averaging because in our experiments it led to higher correlations with the actual WHO grade than the maximum value of the different patches. As previously mentioned, the meningioma training data set consisted of 951 WSIs representing 341 patients and the WHO grade was only available on a patient level for the most malignant tumor of this patient. Hence, the malignancy assigned for a patient was not necessarily consistent with the actual malignancy of the different WSIs. To reduce the resulting label noise, only the WSI with the highest mitotic count per patient was included in the training data set. This reflects the clinical grading workflow, in which also the most malignant tumor specimen is decisive for the overall grade. Therefore, only 341 WSIs of the meningioma grading training dataset where used in training (one per patient, identified by the highest predicted MC). On these slides we performed a 85% / 15% split into training and validation data.

The second approach, denoted as model 2, relies solely on the MC calculated in stage one to predict tumor malignancy. To learn a regression function based on the MC, we used a $1 \times 1$ layer with sigmoid activation followed by a $1 \times 1$ linear layer. The sigmoid activation of the first layer was used to constrain the value range of this layer and thus, to increase the interpretability of the model. Furthermore, the sigmoid function dampens the effects of outliers in the mitotic count on the learned regression function. Overall, this results in a logistic function that can be expressed as

$$t = \text{sig}\left(w_1 \cdot \text{MC} + b_1\right) \cdot w_2 + b_2$$

where $t$ represents the continuous tumor malignancy score, sig() represents the sigmoid function, and $w_1, w_2$ and $b_1, b_2$ the model weights and biases, respectively. Effectively, this results in a scaled and shifted version of the sigmoid function, which is motivated by the biological behavior of tumors: For MC values below a certain cutoff, we can assume normal (regulated) cell division processes; then, there is an intermediate range which scales with the malignancy of the tumor, and above a certain MC value the tumor can be considered so aggressive that we assign the highest grade. This is also in line with the grading schemes for many tumors (e.g., Elston and Ellis (1991); Louis et al. (2016)). As in the first approach, only the highest MC per patient was used for training. We used the same training and

validation split as in approach one.

The third approach, denoted as model 3, uses a combination of the previous two. As in the first approach, it uses a ResNet18 stem to encode morphological information about the patches sampled from the ROI. In addition, the network from model 2 is used to calculate a tumor malignancy score directly from the MC. Both information are merged in a $2 \times 1$ linear layer, which outputs a malignancy score. Like in approach one, the mean of all patches sampled from one ROI is taken as the final malignancy score. The aim of this method is to model the procedure for determining the WHO grade as described by the WHO (Louis et al. (2016)), incorporating both morphological features (as can be extracted by the image-based approach) and the MC.

To ensure the comparability of the results of all three methods, they were trained with the same randomly selected training and validation sub sets. To investigate training robustness we trained all models five times, using different random picks of patients within train and validation set. All models were trained until validation loss converged and the best model was retrospectively selected by the lowest validation loss.

To evaluate the models, we compute both Pearson's correlation and Spearman's correlation between the prediction and the label given by the medical experts. Further, we compute the mean squared error of this prediction over the independent test set. Additionally we computed the portion of correct predictions. To do this, we rounded the predicted tumor malignancy score the the closest integer value and examined how closely the quantized predictions matched the associated WHO grades. We have made our code publicly available at `https://github.com/JonaGanz/automatic_meningioma_grading`.

## 4. Results

We find a generally satisfactory correlation between the inputs and the experts' labels. The models that utilize the mitotic count are generally outperforming the model that only makes use of morphological information from image patches (see Table 1). Overall, we found only a minor impact of the selection of patients as training or validation set, as shown by the low standard deviation across the trained models on the test set. The results of the combined model are on par with the simple logistic regression model only utilizing the mitotic count. This also coincides with low model activation values for the image path in the combined model and thus a generally low impact of the image path on the final model output (cf. Figure 2). Further, we see that the logistic regression model was optimized to yield a rounded grade of one for an MC below approximately four, of two until an MC of approximately 15, and three for MC values above 15 (see Figure 3 right).

## 5. Discussion

The results indicate that the relatively simple model 2 yields the same results as the much more complex combined approach of model 3. This raises the question why model 3 does not give better results although it seems to have more information available. The idea of combining the information of model 1 and model 2 is that we have two independent variables, morphological features and MC, which are both correlated with the WHO grade. We can assume such a correlation since both features are mentioned in the WHO grading

|  | Spearman Correlation | | Pearson Correlation | | Mean Squared Error | | Correct Prediction | |
|---|---|---|---|---|---|---|---|---|
|  | M | SD | M | SD | M | SD | M | SD |
| Model 1 (image-based) | 0.7163 | 0.0178 | 0.7120 | 0.0110 | 0.4161 | 0.1018 | 25/43 | 2.5612 |
| Model 2 (MC-based) | 0.7926 | 0.0004 | 0.7611 | 0.0012 | 0.2416 | 0.0007 | 32/43 | 1.8330 |
| Model 3 (combined) | 0.7900 | 0.0034 | 0.7640 | 0.0025 | 0.2416 | 0.0026 | 33/43 | 0.8000 |

Table 1: Results indicating the mean (M) and standard deviation (SD) of five runs with different training/validation selections. The MC-based regression yields results on par with the more complex combined approach.

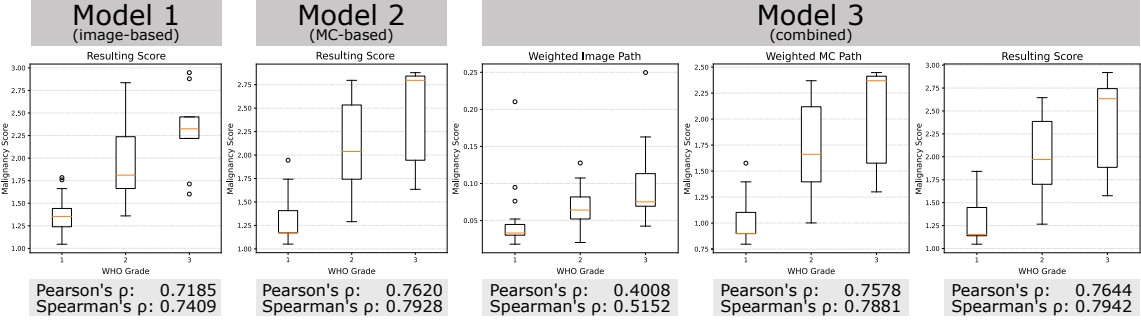

Figure 2: Results for all three approaches, trained using the same training/validation split of patients. For the third (combined) model, the weighted results of both paths are evaluated. Boxes indicate show first and third quartile, whiskers are limited to 1.5 times the interquartile range.

scheme (Louis et al. (2016)). In fact, the results of model 1 and model 2 reveal that these features have discriminative power with respect to the WHO grade. However, just because these two features are univariately correlated with the WHO grade, this does not necessarily mean that their multivariate correlation with the WHO grade is higher. Figure 2 shows the results of all three approaches. Additionally it shows the weighted results for the MC and image path of model 3. It can be seen that the final result of model 3 is mostly determined by the MC path. Although the outputs of the image path are also correlated with the WHO grade, they are weighted so low that their influence on the overall result is minor. This suggests that the image path did not contribute any major additional information to the overall result.

The high discriminative power of the MC with respect to the WHO grade (compared to morphological features retrieved from the image) could be related to the fact that it is the only hard criterion defined in the WHO grading scheme. The regression function between MC and WHO grade learned by model 2 is shown in figure 3. Between the WHO grades 1 and 2, the relationship between MC and WHO grade is almost linear. Between the WHO grades 2 and 3, it shows a logarithmic behaviour and converges against grade 3. This corresponds almost to the different mitotic count thresholds defined in the WHO grading scheme. Our experiments thus indicate that the main driver of the decision for the WHO grade is the mitotic count whereas additional morphological features play a subordinate

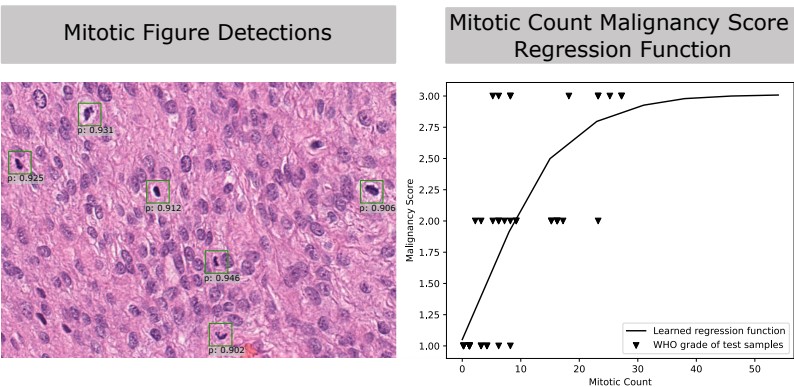

Figure 3: Left panel: exemplary detections of mitotic figures by the Faster RCNN approach in Stage 1. Right panel: Logistic regression learned by the model 2.

role, at least for our data set.

Another limitation of our experiment is that our test only contained 141 slides from 43 patients. However, since we also found a good correspondence of the results to each of our validation runs, we are confident that our observations can generalize also for larger data sets. At the same time, a generalization of these findings and suggesting that morphological features can be regarded as negligible would be premature. The malignancy of a tumor is a continuous biological parameter for which a discrete value like the WHO grade can only be an approximation. Future work should therefore assess the proposed model variants on a more continuous grading scheme for malignancy, the derivation of biological parameters like genotypical information or prediction of risk of recurrence, for which a more pronounced impact of morphological features is likely.

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
