# OpenReview forum: "Automatic and explainable grading of meningiomas from histopathology images"
_MICCAI.org/2021/Workshop/COMPAY — COMPAY 2021_

### Official Review · Reviewer_HCYi · 2021-08-09
**An innovative, multimodal approach to classifying meningiomas**

**Rating:** 9
**Confidence:** 4

**Review:**

The authors present an innovative approach to predicting meningioma grades from mitotic count as well as morphological features from image patches. Their approach enables the authors to assess the relative importance of these two input factors, yielding some insight in the importance of mitotic count as a "hard factor" in glioma grading. The study is well designed and the paper clearly written, and seems well suited for COMPAY.

Two issues should be addressed for the final version of the contribution:

- As for data-driven models, class imbalance is key for the performance assessment of the trained model. The authors should therefore include the distribution of WHO grades in their training and test cohorts to characterize their dataset more completely.

- From the current manuscript, the model seems to be built on a monocentric dataset. Have the authors tried to assess its generalizability on external data? If not, which type of external validation would they suggest?

Additionally, the authors might consider two additional points:
- Have the authors condidered applying explainability methods (e.g., saliency maps, Grad-CAM or LRP) to their morphology patches to test if the model focusses on relevant regions of the images?
- For model 2 and 3, the authors state that "the mean of all patches sampled from one ROI is taken as the final malignancy score".
What is the motivation of this choice? Should the overall grade not correspond to the maximum score incountered in a ROI? Please specify.

---

### Official Review · Reviewer_cW8k · 2021-08-22
**Explainable grading of meningiomas**

**Rating:** 5
**Confidence:** 5

**Review:**

Overall, the paper is well written and the aims of the paper are very clear. The paper aims to build an automatic method for meningioma grade classification that uses both image-based features and the mitotic count.

The main limitation of the work is the technical contribution. For object detection, the authors use Faster-RCNN, which is a widely used technique for object detection, and a simple strategy for utilising image based features. Therefore, my opinion is that the main contribution of this work is the well conducted experiments that conclude that the mitotic count is predictive of the grade of meningioma. Despite this, as the authors mention, this is already used for routine grading of meningioma cases and therefore is not new. However, this study provides proof that machine learning models provide promise into implementing such models in clinical practice.

The work mentions that a 'pretrained ResNet' was used to extract image-based features. Can the authors provide further details into what it was trained on? For example, were ImageNet features used? Do you think you could improve the results by using a pre-trained model on digital pathology (DPath) images? For example, contrastive learning could be used to extract DPath specific features. Also, were the encoder weights frozen or did you train all of the weights? Also, the model will only be explainable if only the mitotic count is used in the model - the deep features will be very hard to interpret. It would be interesting to instead perform nuclear segmentation and extract some interpretable nuclei-based features to make the entire pipeline explainable. This repository below could be used to perform nuclear segmentation:

https://github.com/vqdang/hover_net

Overall, the paper is well written and the experiments are good, but I question the contribution of the work.

---

### Decision · Program_Chairs · 2021-08-25

Accept